# Detection of Ultra-Rare *ESR1* Mutations in Primary Breast Cancer Using LNA-Clamp ddPCR

**DOI:** 10.3390/cancers15092632

**Published:** 2023-05-06

**Authors:** Yoko Hashimoto, Nanae Masunaga, Naofumi Kagara, Kaori Abe, Tetsuhiro Yoshinami, Masami Tsukabe, Yoshiaki Sota, Tomohiro Miyake, Tomonori Tanei, Masafumi Shimoda, Kenzo Shimazu

**Affiliations:** 1Department of Breast and Endocrine Surgery, Graduate School of Medicine, Osaka University, 2-2-E10 Yamadaoka, Suita 565-0871, Osaka, Japan; 2Department of Breast Surgery, Osaka General Medical Center, 3-1-56, Bandai-Higashi, Sumiyoshi-ku, Osaka 558-8558, Osaka, Japan

**Keywords:** *ESR1* mutation, primary breast cancer, minor clone, clamping PCR, droplet dPCR

## Abstract

**Simple Summary:**

*ESR1* mutations in breast cancer are one of the mechanisms of resistance to aromatase inhibitors (AI). These mutations are common in metastatic breast cancer (MBC). In past reports, mutations in primary tumors were so rare that they were thought to occur de novo with AI therapy. Conversely, previous studies using droplet digital polymerase chain reaction (ddPCR) have suggested the existence of *ESR1*-mutant minor clones with low allele frequencies; however, no large-scale studies have been conducted. In this study, we attempted to detect ultra-rare *ESR1* mutations in primary breast cancer tumors using locked nucleic acid (LNA)-clamp ddPCR, and 28 *ESR1* mutations were found in 27 patients. Most of these mutations were minor clones with a variant allele frequency of <0.1% which may have been overlooked by conventional methods. LNA-clamp ddPCR can also be applied to detect other gene mutations, which would be very useful.

**Abstract:**

*ESR1* mutations in breast cancer are one of the mechanisms of resistance to aromatase inhibitors. These mutations are common in metastatic breast cancer; however, these are rare in primary breast cancer. However, these data have been analyzed mainly in formalin-fixed, paraffin-embedded tissue; thus, rare mutations that may be present in primary breast cancer may be overlooked. In this study, we developed a highly sensitive mutation detection method called locked nucleic acid (LNA)-clamp droplet digital PCR (ddPCR) and validated it. The mutation detection sensitivity was substantiated to 0.003%. Then, we used this method to analyze *ESR1* mutations in fresh-frozen (FF) tissues of primary breast cancer. cDNA extracted from the FF tissues of 212 patients with primary breast cancers were measured. Twenty-eight *ESR1* mutations were found in twenty-seven (12.7%) patients. Sixteen (7.5%) patients had Y537S mutations and twelve (5.7%) had D538G mutations. Two mutations with a variant allele frequency (VAF) of ≥0.1% and twenty-six mutations with a VAF of <0.1% were found. By using this LNA-clamp ddPCR, this study demonstrated the presence of minor clones with a VAF of <0.1% in primary breast cancer.

## 1. Introduction

Approximately 70% of breast cancers are estrogen receptor (ER)-positive and are eligible for endocrine therapy [1]. Endocrine therapy, which has fewer side effects, is generally the preferred treatment for ER-positive breast cancer [2,3], although many cases develop resistance during treatment [4]. Recently, mutations in the *ESR1* gene, which encodes for the estrogen receptor (ER), have been reported as one mechanism of resistance and are found to occur frequently after long-term aromatase inhibitor (AI) therapy [5]. These mutations result in estrogen-independent ER activity [6,7,8], which is resistant to various subsequent therapies and affects overall survival [9,10]. The expression of *ESR1* mutations in breast cancer is more common in hormone receptor-positive metastatic breast cancers (MBC) [6,7,11,12,13,14]. On the contrary, *ESR1* mutations in primary breast cancer have been reported to be extremely rare [6,13,15,16]. Thus, *ESR1* mutations have been thought to occur with aromatase inhibitor (AI) therapy [14]; however, these *ESR1* mutations in primary tumors have been analyzed mainly through formalin-fixed paraffin-embedded (FFPE) tissue [15,17]. The mutation detection sensitivity using FFPE is approximately 10% due to DNA denaturation caused by deamination and other reactions [18,19]. However, recent studies have suggested that *ESR1* mutations with allele frequencies of ≤10% may exist by using highly sensitive detection methods such as ddPCR [20,21]. Thus, we hypothesized that *ESR1*-mutant subclones present in trace amounts in primary tumors may selectively proliferate under low estrogen levels in AI therapy, leading to recurrence. Based on this hypothesis, we attempted to detect trace amounts of *ESR1* mutations using fresh-frozen (FF) tissues of primary tumors which may have been missed in the past.

To detect minor clones in primary tumors, a more sensitive detection method was needed. The theoretical sensitivity of droplet digital polymerase chain reaction (ddPCR) was reported to be 0.001% [11,22]. However, the maximum amount of DNA that can be submitted at one time is 20,000 copies. Usually, two or more dots above the threshold are considered mutant-positive, so the actual sensitivity for mutant detection is approximately 0.01% [23]. The sensitivity of next-generation sequencing (NGS) is approximately 0.05–2% [23,24,25]. However, in clinical samples, it is approximately 0.1% [26]. Therefore, we developed a high-sensitivity mutation detection method called LNA-clamp ddPCR. LNA-clamp ddPCR is a PCR method that can amplify only the target sequence by mixing in LNA, which is an artificial nucleic acid that specifically binds to a specific sequence [27,28]. When PCR is performed with a mixture of LNA that specifically binds to the wild-type sequence, the amplification of the wild type is inhibited and only the mutant type is amplified (Appendix A). This method can increase the MAFs and be used to detect rare mutations that cannot be detected via conventional methods. In addition, mRNA was used instead of DNA to obtain numerous *ESR1* copies and detect pathogenic mutations with gene expression.

In this study, we used this method to analyze *ESR1* mutations in mRNA extracted from FF tissues of 212 ER-positive primary breast cancers. The *ESR1* mutations were analyzed for the two most representative locations, Y537S (1610A>C) and D538G (1613A>G). These mutations are the most frequent sites of *ESR1* mutations [16,29]. The mutations detected were differentiated into those that could be detected with the sensitivity of conventional methods (defined as major clones) and those that required more sensitivity (defined as minor clones).

## 2. Materials and Methods

### 2.1. Cell Culture and DNA/mRNA Extraction

Ten breast cancer cell lines (MCF7, T47D, ZR75-1, ZR75-30, BT483, BT20, MDA-MB-231, SK-BR3, MDA-MB-468, and MDA-MB-453) obtained from the American Type Culture Collection (Manassas, VA, USA) were cultured according to the culture guidelines of the American Type Culture Collection. Equal amounts of DNA and mRNA were extracted from the same dish cell-suspended liquid using the DNeasy Blood and Tissue Kit and the RNeasy Mini Kit (Qiagen, Hilden, Germany) according to the manufacturer’s instructions (https://www.qiagen.com/us/products/discovery-and-translational-research/dna-rna-purification/dna-purification/genomic-dna/dneasy-blood-and-tissue-kit (accessed on 20 March 2023), https://www.qiagen.com/us/products/discovery-and-translational-research/dna-rna-purification/rna-purification/total-rna/rneasy-kits?catno=74104 (accessed on 20 March 2023)). Then, *ESR1* expression levels in the DNA and mRNA were compared in 10 breast cancer cell lines.

### 2.2. Patients and Samples

This study included a total of 212 patients who were treated at Osaka University Hospital between 2009 and 2018. All patients had estrogen receptor (ER)-positive breast cancer. ER positivity was defined as an Allred score of >3 [30]. The clinicopathological characteristics of the patients are shown in Table 1. FF tissues were obtained from the primary tumor at the time of surgery. No patients received preoperative therapy before tissue sampling.

Informed consent was obtained from all patients before sampling, and the Ethical Review Board of Osaka University Hospital approved this study (Ethics approval No. 20399). This study was conducted with anonymous numbers so that individuals could not be identified.

### 2.3. mRNA Extraction from Primary Tumors and Reverse Transcription

Tumor mRNA was extracted from FF tissues using the RNeasy Mini Kit (Qiagen) according to the manufacturer’s instructions. Tumor mRNA was reverse-transcribed to cDNA using the ReverTra Ace qPCR RT Kit (TOYOBO, Osaka, Japan) according to the manufacturer’s instructions (https://lifescience.toyobo.co.jp/user_data/pdf/products/manual/FSQ-101.pdf, accessed on 20 March 2023). 

### 2.4. LNA-Clamp ddPCR from Primary Tumors

The PCR clamping assay was performed using the specific LNA oligo (Ajinomoto Bio Pharma, Osaka, Japan). The targets of the *ESR1* mutations were Y537S (1610A>C) and D538G (1613A>G). Each of the LNA oligos are shown in Appendix A. Y537S and D538G DNA oligos (FASMAC, Kanagawa, Japan) were used as positive controls. We confirmed that these oligos worked with the primers and probes we used. Each of the mutant oligos are shown in Appendix A. For ddPCR, the QX200 Droplet Digital PCR System (Bio-Rad Laboratories, Hercules, CA, USA) was used.

ddPCR was performed in a 20-μL reaction containing 8.6 μL of template cDNA, 10 μL of ddPCR Supermix (Bio-Rad Laboratories), 0.9 μM each of the forward and reverse primer, 0.9 μM of specific LNA oligo, and 0.25 μM each of the wild-type and mutant (Y537S or D538G) probes. The primers and probes used are shown in Appendix A. Then, 20 μL of PCR reaction and 67 μL of droplet generation oil were loaded into a cartridge of the droplet generator and placed onto a thermal cycler for PCR. The cycling conditions were as follows: one cycle of 95 °C for 10 min, 40 cycles of 94 °C for 30 s and 54 °C for 1 min, and one cycle of 98 °C for 10 min.

As a control, a normal ddPCR without LNA oligo was performed simultaneously. The input cDNA copy number was calculated from this result. Regarding the standardization of the samples, we first checked the copy number of the reverse-transcribed cDNA ddPCR using 1 μL of extracted DNA, and we adjusted the input copy number to 100,000 copies for LNA-clamp ddPCR.

### 2.5. Statistics

R (ver. 4.0.3) was used for statistical processing. To examine the significance of the association, Fisher’s exact test was used to compare 2 × 2 groups, and the Mann–Whitney U test was used to compare the age distribution and duration of therapy. A two-way analysis of variance and Tukey’s test were used to analyze sensitivity. A *p* < 0.05 was considered significant.

## 3. Results

### 3.1. Comparison of ESR1 Expression Levels between DNA and mRNA in Cell Lines

*ESR1* expression levels between DNA and mRNA were compared in 10 breast cancer cell lines to verify that mRNA provides a higher copy number in ER-positive breast cancer. mRNA had higher *ESR1* expression levels than DNA in ER-positive cells (the mRNA/DNA ratios were as follows: MCF7: 8.92, T47D: 3.94, ZR75-1: 1.62, ZR75-30: 1.62, and BT483: 1.02). *ESR1* was less expressed in mRNA than in DNA in ER-negative cells (the mRNA/DNA ratios were as follows: BT20: 0.27, MDA-MB-231: 0.12, SKBR3: 0.01, MDA-MB-468: 0.01, and MDA-MB-453: 0.00) (Appendix A). Therefore, this study targeted mRNA from ER-positive tumors.

### 3.2. Confirmation of the Clamping Method Using LNA Oligo

To confirm the clamping effect of the designed *ESR1* Y537S (1610A>C) LNA oligo and *ESR1* D538G (1613A>G) LNA oligo, PCR clamping was performed on samples with serially diluted wild-type oligo from 6000 to 60,000,000 copies. The concentrations of LNA oligo were 0, 1, and 3 μM. Both the Y537S and D538G LNA oligos could be clamped completely up to 60,000,000 copies at LNA 3 μM (Appendix A).

To examine the effect of LNA oligo on the mixture sample of mutant DNA and wild-type DNA collected from leukocytes of healthy volunteers, mutant DNA (Y537S and D538G) was serially diluted into wild-type DNA (total copy number (mutant + wild-type) = 20,000 copies). Mixed samples of 100%, 33%, 10%, 3.3%, 1.0%, 0.3%, 0.1%, and 0% were prepared. PCR was performed on these DNA samples with and without the LNA oligo to examine the effect of the LNA oligo on mutant DNA. Consequently, the LNA oligo completely blocked the wild-type DNA but did not affect the mutant DNA (Appendix A).

### 3.3. Detection Sensitivity and Cutoff of the LNA-Clamp ddPCR

We examined the effect of the LNA-clamp ddPCR on the analysis of mutant DNA mixed with an excess amount of wild-type DNA. Samples were prepared by mixing 100,000 copies of wild-type DNA from the leukocytes of healthy volunteers with serially diluted mutant DNA (Y537S) (0, 1, 3, 10, 33, and 100 copies) at six concentrations. PCR was performed with and without the LNA oligo to confirm the sensitivity of the clamp method. Moreover, a sample of mutant DNA (Y537S) (0, 1, 3, 10, 33, and 100 copies) mixed with 10,000 copies of wild-type DNA was also prepared as a reference (Figure 1).

A two-way analysis of variance was performed for the three methods of measurements (with LNA, without LNA, and reference) and six concentration samples, and the interaction was observed (*p* < 0.001). Therefore, a multiple-comparison test (Tukey’s test) was performed. The results of Tukey’s test showed that all samples measured using the clamp method (with LNA) did not differ from the reference results (*p* = n.s.). The no-clamp method (without LNA) was found to have inferior power compared to the reference results for samples from 0.1% to 0.003% (*p* < 0.05).

Then, LNA-clamp ddPCR was performed on Y537S and D538G using cDNA reverse-transcribed from mRNA extracted from 26 normal mammary tissues and the leukocytes of 19 healthy volunteers to measure the background amplitude of the LNA-clamp ddPCR (Appendix A). For each well, the fluorescence intensities of the highest and second-highest dots were calculated and a cutoff line was set so that no well had more than two positive dots in the background sample. Consequently, the threshold lines were set at a fluorescence intensity of 1400 for Y537S and 2250 for D538G.

### 3.4. ESR1 Mutation Analysis of Primary Breast Cancer Using LNA-Clamp ddPCR

*ESR1* mutations were measured with an LNA-clamp ddPCR of cDNA extracted from FF tissues of 212 patients with primary breast cancers. Twenty-eight *ESR1* mutations were found in twenty-seven (12.7%) patients (Figure 2), which included sixteen (7.5%) Y537S mutations and twelve (5.7%) D538G mutations. Two mutations with a VAF of ≥0.1% (as major clones) and twenty-six mutations with a VAF of <0.1% (as minor clones) were found. Double mutation was observed in one case. Mutations and VAF for each case are shown in Table 2. All cases underwent conventional ddPCR before LNA-clamp PCR, but only major clones were detected, not minor clones. Thirty cases also underwent additional NGS as a pilot study, but no minor clones were detected. No relevance was found between clinicopathological features and mutation status (Table 3). The details of postoperative AI treatment is shown in Appendix A. Regarding only the patients with recurrence, the *ESR1* mutation-positive group tended to have a shorter duration of AI treatment for recurrence, but this was not statistically significant (Appendix A).

## 4. Discussion

In this study, we established the LNA-clamp ddPCR to detect *ESR1* minor clones with high sensitivity. The clamp method is a PCR method in which only the target base sequence can be amplified by mixing artificial nucleic acids (LNA oligo, etc.) that bind specifically to a particular base sequence [27,31,32]. Then, we developed a method to detect trace amounts of mutant DNA by specifically clamping *ESR1* wild-type DNA using LNA oligo. Furthermore, mRNA was used instead of DNA to obtain a larger copy number of *ESR1* and detect pathogenic mutations with gene expression.

We demonstrated the presence of minor clones with a VAF of <0.1% in primary breast cancer using this LNA-clamp ddPCR. Then, 212 primary breast cancers were analyzed, find that 27 patients (12.7%) had 28 *ESR1* mutations. Twenty-six mutations were minor clones with a VAF of <0.1% which conventional methods (NGS, MB-NGS, and ddPCR) may have overlooked.

We used FF tissues of the primary tumor instead of FFPE to detect mutations, and this is a very valuable finding. In this study, Y537S and D538G mutations were present in ligand-binding domain hotspots and accounted for approximately 50% of all *ESR1* mutations [6,11,33]. Y537S and D538G mutations were found in approximately 12.7% of the primary tumors analyzed in this study, and approximately 25% of *ESR1* mutations of whole exons were present in the primary tumors, which is close to the frequency of *ESR1* mutations reported for recurrent breast cancers (20–50%) [7,11,14,33,34,35]. These results support our hypothesis.

In this study, no significant difference was found in prognosis according to the presence or absence of *ESR1* mutations in primary tumors. One of the reasons for this is that only 12 (44%) of the 27 patients with *ESR1* mutations in primary tumors received AI as adjuvant therapy. Even if the primary tumor has an *ESR1* mutation as a minor clone, it has no effect on the prognosis when clonal selection could be avoided. When we discussed only 13 recurrent cases, the *ESR1* mutation-positive group tended to have a shorter response time to AI treatment for recurrence (not statistically significant). This suggests resistance to AI due to clonal selection, but these results are based on very few cases. A larger number of cases would strengthen the significance of this result.

As for study limitations, first, only two *ESR1* hotspot mutations (Y537S and D538G) were analyzed. We focused on two mutations with a higher frequency of *ESR1* mutations to extract and analyze the maximum amount of mRNA from a limited amount of tumor tissue because ddPCR requires analysis for each position of the mutation. Second, because of the limited number of cases that recurred, it was not possible to trace the mutations found in the primary tumor in the recurrent tissue.

In the future, we plan to analyze other *ESR1* mutations using LNA-clamp ddPCR to determine the mechanism of recurrence within *ESR1* mutations. We also plan to apply this method to analyze other gene mutations.

## 5. Conclusions

We have developed a highly sensitive mutation detection method called LNA-clamp ddPCR, and this study demonstrated the presence of minor clones with a VAF of <0.1% in primary breast cancer.

## Figures and Tables

**Figure 1 cancers-15-02632-f001:**
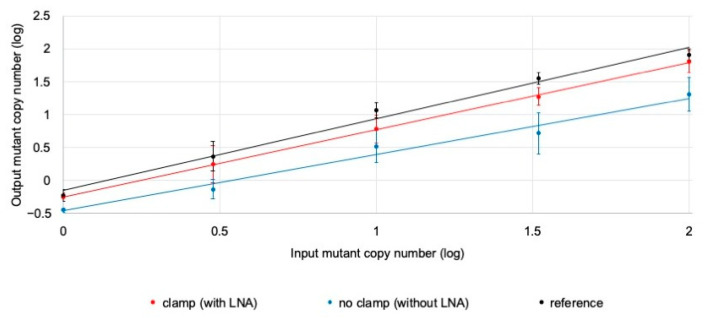
Sensitivity test for LNA-clamp PCR. Comparison of the sensitivity with serially diluted Y537S mutant DNA (1, 3, 10, 33, and 100 copies) in wild-type DNA (with or without clamp; the references are 100,000 copies and 10,000 copies, respectively). The input mutant copy number (log) was plotted on the X-axis and the output mutant copy number (log) was plotted on the Y-axis. The samples with the LNA oligo (clamp: red line), without the LNA oligo (no clamp: blue line), and the reference (black line) are shown. Error bars indicate the standard deviation over 10 experiments.

**Figure 2 cancers-15-02632-f002:**
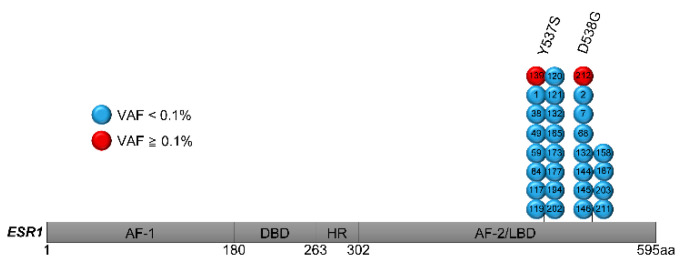
Schematic distribution of *ESR1* mutations identified with LNA-clamp ddPCR. FF tissue from 212 primary tumors was analyzed with LNA-clamp ddPCR. Twenty-six mutations with VAF < 0.1% (blue) and two mutations with VAF ≥ 0.1% (red) were detected in twenty-seven patients. The number in each lollipop represents the patient number. *AF-1*—activation function-1, *DBD*—DNA-binding domain, *HR*—hinge region, *AF-2*—activation function-2, and *LBD*—ligand-binding domain.

**Table 1 cancers-15-02632-t001:** Clinicopathological characteristics of the patients.

N		212
Age	Median (range)	60 (33–96)
Tumor size (cm)	≤2	168
	>2	44
Lymph node metastasis	Negative	178
	Positive	34
Histological grade	1/2	184
	3	28
ER	Positive	212
	Negative	0
PgR	Positive	177
	Negative	35
HER2	Positive	26
	Negative	186
Histology	IDC	174
	ILC	17
	Other	21
Prognosis	No recurrence	199
	Recurrence	13

ER—estrogen receptor, HER2—human epidermal growth factor receptor 2, IDC—invasive ductal carcinoma, ILC—invasive lobular carcinoma, and PgR—progesterone receptor.

**Table 2 cancers-15-02632-t002:** Detailed clamping PCR results for 28 *ESR1* mutations.

Case	AA Change	SNV	VAF (%)	Mutant Copy Number (per Well)	Total Copy Number (per Well)
1	Y537S	1610A>C	0.0025	2.8	111,150
38	Y537S	1610A>C	0.0031	3.6	116,800
49	Y537S	1610A>C	0.0022	2.8	129,000
59	Y537S	1610A>C	0.0028	3	108,800
84	Y537S	1610A>C	0.0018	2.8	158,000
117	Y537S	1610A>C	0.0023	2.4	106,540
119	Y537S	1610A>C	0.0021	2.6	121,200
120	Y537S	1610A>C	0.0024	2.6	108,500
121	Y537S	1610A>C	0.0037	3.8	102,200
139	Y537S	1610A>C	0.1580	6.8	4304
165	Y537S	1610A>C	0.0053	2.6	48,640
173	Y537S	1610A>C	0.0106	3	28,320
177	Y537S	1610A>C	0.0025	2.8	112,800
194	Y537S	1610A>C	0.0026	2.6	101,700
202	Y537S	1610A>C	0.0028	2.6	91,280
132	Y537S	1610A>C	0.0034	2.8	81,280
132	D538G	1613A>G	0.0034	2.8	81,280
2	D538G	1613A>G	0.0041	2.6	63,840
7	D538G	1613A>G	0.0119	2.8	23,520
68	D538G	1613A>G	0.0110	4.8	43,520
144	D538G	1613A>G	0.0422	2.8	6640
145	D538G	1613A>G	0.0060	2.8	46,690
146	D538G	1613A>G	0.0111	3	27,040
158	D538G	1613A>G	0.0053	3	56,480
187	D538G	1613A>G	0.0053	2.8	52,800
207	D538G	1613A>G	0.0111	2.6	23,520
211	D538G	1613A>G	0.0039	3.8	96,480
212	D538G	1613A>G	1.6260	36	2178

AA—amino acid, SNV—single nucleotide variant, and VAF—variant allele frequency.

**Table 3 cancers-15-02632-t003:** Clinicopathological features of patients and comparison with and without *ESR1* mutation.

		*ESR1*		*p*
		Positive	Negative	
N		27	185	
Age	Median (range)	60 (36–83)	60 (33–95)	0.32 *
Tumor size (cm)	≤2	23	145	0.61 **
	>2	4	40	
Lymph node metastasis	Positive	4	30	1.00 **
	Negative	23	155	
PgR	Positive	23	154	1.00 **
	Negative	4	31	
HER2	Positive	2	24	0.54 **
	Negative	25	161	
Histological grade	1	11	77	1.00 **
	2/3	16	108	
Prognosis	No recurrence	24	175	0.22 **
	Recurrence	3	10	

HER2—Human epidermal growth factor receptor 2, PgR—progesterone receptor. * Mann–Whitney U test, ** Fisher’s exact test.

## Data Availability

Data is contained within the article or Appendix A.

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
