# Peer review of "Detection of Ultra-Rare ESR1 Mutations in Primary Breast Cancer Using LNA-Clamp ddPCR"

_cancers, 2023, doi:10.3390/cancers15092632_

Round 1

Reviewer 1 Report

The authors describe a new PCR technique (LNA clamp ddPCR) to detect low copy mutation ESR1 in primary breast tumor. However, major revisions are required before publication of the manuscript.

Comments:

1. Supplementary figures S1-S4 described in text are missing from the manuscript. Without validation data, this paper cannot be interpreted.

2. There is no explanation of LNA-clamp ddPCR technique in methods or introduction. What is the novelty over tradition PCR? LNA - clamp ddPCR method needs to be explained properly preferably with a schematic figure.

3. To show relevance of this method, traditional PCR needs to be performed side by side with LNA-clamp PCR in Table 2 in all the patients to justify the sensitivity of this technique.

Without these information, the paper is not comprehensible or suitable for publication

Author Response

Reviewer1

Comment #1. Supplementary figures S1-S4 described in text are missing from the manuscript. Without validation data, this paper cannot be interpreted.

Response:

We apologize for forgetting to attach the supplementary figures. We attached Supplementary figures.

Comment #2. There is no explanation of LNA-clamp ddPCR technique in methods or introduction. What is the novelty over tradition PCR? LNA - clamp ddPCR method needs to be explained properly preferably with a schematic figure.

Response:

LNA-clamp ddPCR is a PCR method that can amplify only the target sequence by mixing LNA which is an artificial nucleic acid and specifically binds to a specific sequence. When PCR is performed with a mixture of LNA that specifically binds to the wild type sequence, the amplification of the wild type is inhibited and only the mutant type is amplified. We added a figure that schematizes the LNA-clamp ddPCR as Figure S1. This method can increase the MAFs and be used to detect rare mutations that cannot be detected by conventional methods. This has been described in the Introduction (line 66 on page 2 – line 72 on page 2).

Comment #3. To show relevance of this method, traditional PCR needs to be performed side by side with LNA-clamp PCR in Table 2 in all the patients to justify the sensitivity of this technique.

Response:

All cases underwent conventional ddPCR before LNA-clamp PCR, but only major clone was detected, not minor clones. Thirty cases also underwent additional NGS as a pilot study, but also, no minor clones were detected.) This has been described in the Results (line 200 on page 5 – line 203 on page 5).

Reviewer 2 Report

Detection of Ultra-Rare ESR1 Mutations in Primary Breast Cancer Using LNA-clamp ddPCR 

In this study, we attempted to detect ultra-rare ESR1 mutations in primary breast cancer tumours by the LNA-clamp ddPCR, and 28 ESR1 mutations were found in patients. Please refine the acronym LNA.

The authors find 28 ESR1 mutations in 27 primary BC samples and classify them.

The authors should explain and link method sensitivity with allele frequency in lines 42-45.

Why is this manuscript so brief? Is it in a short communication format? If not, the introduction requires more detail about BC, subtypes, treatment, resistance to treatment, and information on various mutations. The authors have rapidly got to the point without enough momentum building. 

The authors hypothesise that some ESR1 mutations that lead to aromatase resistance are truncal and are selected for under low treatment concentrations.

The theoretical sensitivity of droplet digital polymerase chain reaction (ddPCR) was reported to be 0.001% [14,2]. However, the maximum amount of DNA that can be submitted at one time is 20,000 copies, and the actual sensitivity for detection is approximately 0.01%. The authors could explain more about the differences between theoretical and actual sensitivity and link this to the copy numbers required.

I am not sure why the authors feel it is necessary to talk about the sensitivity of other methods such as NGS and MB-NGS if they are not using these methods in their short manuscript.

However, in clinical samples, it is approximately 0.1%. What does this figure (0.1%) refer to?

Again, if this is not a short format article, it could do with more introduction.

Methods:

DNA/RNA was extracted from BC cell lines.

Also, FFPE samples from 212 patients have been utilised. ER+ BC has been mentioned here but this has not been sufficiently characterised and introduced in the introduction section.

Again, the authors assume reader knowledge of various hormone receptor statuses in table 1.

Ethical approval information has been provided which is great.

2.3. mRNA extraction from FFPE

Why did the authors only test these mutations Y537S (1610A>C) and D538G (1613A>G)? Again, the introduction did not sufficiently introduce rare and common mutations of ESR1 and evidence supporting the reported frequency of any of these.

Results:

The ESR1 expression mRNA/ DNA was measured in 10 cell lines, please explain in more detail why this ratio was necessary to establish.

3.2.

Please check your supplementary files since I can only see a file with oligo/ primer information and cannot see figure Ss and have not been able to evaluate it.

3.3.

The authors could add some content on the LNA-clamp method. Also, how was their positive control (DNA of Y537S mutation) established? Where was this known DNA obtained from and how was it validated? Also, was the other mutation also validated using this standardised method? D538G? If not, why not? If possible please add this information.

Figure 1. Sensitivity test of LNA-clamp PCR. Comparison of the sensitivity with serially diluted Y537S mutant DNA (1, 3, 10, 33, and 100 copies) in wild-type DNA (clamp and no clamp, 100,000 copies; reference, 10,000 copies).  The latter bracket is confusing please revise the wording and use “respectively” where appropriate so the various copy numbers of each condition are easier to follow.

Figure 1, there is a steady and linear increase in mutation detection following increasing levels of DNA copies which is reassuring. Also, the clamp method is more robust than no clamp.

Again, I can’t check Figure S4 since they don’t appear in my downloaded supplementary files, the title below is the only thing that I can see.

Table S1. Primers, probes, and LNA-oligos for ddPCR. 

In lines 160-162, can the authors provide information on between patient DNA copy normalisation? How did the authors standardise their samples for various patients? From Table 3 I can see that different copy numbers were used for each patient. Were any efforts made to dilute each patient sample and measure VAFs to see if the percentage holds?

3.4. 

In lines 169-171, the authors have provided the count of patient samples showing the two mutations and have calculated the VAF in the associated table which is good.

I was going to suggest that the authors add indicate details about the 212-patient mutation status but since only 2 mutations were studied, table 3 summarises this information. However, later in the manuscript, the authors talk about therapy status so you could give more information about this in a table.

The authors have performed statistical analysis to compare the clinicopathological features between the larger group 212 and those of the 2-ESR1 mutation group. What type of characteristics has been enriched in the latter group (please add this information to the results section in line 174)? The authors haven’t explained this finding at all. Did any of the patients show both mutations, if not why?

It would be useful if the authors could draw some comparisons with patients' post-aromatase therapies and if/how the VAFs of these two mutations have been altered. This will give some preliminary clues to the truncal/ clonal evolution of these genetic alterations and how treatment may affect them. This could be a comparison to other published studies. The authors could easily perform some statistical analysis to compare them. This is particularly a question since they mention this point in their introduction.

Also, with respect to the other patients who did not show these 2 mutations, was this a copy number/ sensitivity issue rather than a 0/1 absent/present issue? Studies have suggested that many primary tumours carry resistance genes in low copies (the point the authors refer to). If so, what method do the authors suggest the scientific community could use to find these ultra-low copy mutations?

Y537S and D538G mutations were found in approximately 12.7% of the primary tumors analyzed in this study, and approximately 25% of ESR1 mutations of whole exons are present in primary tumors, which is close to the frequency of ESR1 mutations reported for recurrent breast cancers (20%–50%) [2,3,5,23–25]. These results support our hypothesis.  Yes, this is the point I was making earlier. The authors could elaborate on the significance of this finding.

In this study, no significant difference was found in prognosis according to the presence or absence of ESR1 mutations. I assume the authors refer to the bottom of Table 3. I am confused about what they mean since the p-value is significant.

This is thought that only 12 (44%) of the 27 patients with ESR1 mutations received AI as adjuvant therapy.  This is news to me since I thought that all the patients were pre-treatment and this was not discussed in the manuscript. Please add sections to the manuscript to outline pre-treatment and post-treatment patients in this study and add plots to show the distinction between these two groups with respect to treatment status also update the tables with this information.  Also, what is meant by it is thought? Treatment status is not a matter of contention, 12/27 patients have either received treatment or they haven’t, kindly alter this sentence.

Second, recurrent cases were limited; thus, we could not confirm the same ESR1 mutation in recurrent tissues. The authors should give more detail about which samples were recurrent. I thought all samples used were primary (primary meaning primary tumour as opposed to patient tissue). How many of your samples represented recurrent disease or secondary tumours?

From the conclusion I am not sure if the authors felt that their manuscript is more of a methods article or a patient mutation article.

Author Response

Reviewer 2

In this study, we attempted to detect ultra-rare ESR1 mutations in primary breast cancer tumours by the LNA-clamp ddPCR, and 28 ESR1 mutations were found in patients.

Comment #1. Please refine the acronym LNA.

Response:

LNA means locked nucleic acid.

This has been described in the Simple Summary (line 17 on page 1) and Abstract (line 25-26 on page 1).

Comment #2. The authors should explain and link method sensitivity with allele frequency in lines 42-45.

Response:

The sensitivity of mutation detection using FFPE is approximately 10%. However, recent studies have suggested that ESR1 mutations with allele frequencies of ≤10% may exist by using highly sensitive detection methods such as ddPCR. We added the text in the Introduction (line 53-54 on page 2).

Comment #3. Why is this manuscript so brief? Is it in a short communication format? If not, the introduction requires more detail about BC, subtypes, treatment, resistance to treatment, and information on various mutations. The authors have rapidly got to the point without enough momentum building. 

Response:

Thank you for your suggestion.

We added the following text at the beginning of the Introduction (line 37 on page 1 - line 45 on page 2).

Approximately 70% of breast cancers are estrogen receptor (ER)-positive and are eligible for endocrine therapy. Endocrine therapy with fewer side effects is generally the preferred treatment for ER-positive breast cancer, although many cases develop resistance during treatment. Recently, mutations in the ESR1 gene, which encodes for the estrogen receptor (ER), have been reported as one mechanism of resistance and are found to occur frequently after long-term aromatase inhibitor (AI) therapy. These mutations result in estrogen-independent ER activity, which is resistant to various subsequent therapies and affects overall survival.

Comment #4. The theoretical sensitivity of droplet digital polymerase chain reaction (ddPCR) was reported to be 0.001% [14,2]. However, the maximum amount of DNA that can be submitted at one time is 20,000 copies, and the actual sensitivity for detection is approximately 0.01%. The authors could explain more about the differences between theoretical and actual sensitivity and link this to the copy numbers required.

Response:

The maximum amount of DNA that can be submitted at one time by ddPCR is 20,000 copies. Usually, two or more dots above the threshold are considered mutant positive, so the actual sensitivity for mutant detection is approximately 0.01%. We added the text in the Introduction (line 61 on page 2 - line 64 on page 2).

Comment #5. I am not sure why the authors feel it is necessary to talk about the sensitivity of other methods such as NGS and MB-NGS if they are not using these methods in their short manuscript.

Response:

We apologize for not writing about that point. We compared with major mutation analysis methods to LNA-clamp ddPCR. The sensitivity of NGS and MB-NGS in clinical samples is approximately 0.1%. We thought this would not meet the required sensitivity. Thirty cases were analyzed by MB-NGS before LNA-clamp ddPCR as a pilot study. But only major clone was detected, not minor clones. The results of the NGS analysis were added in the Results (line 202 on page 5 - line 203 on page 5).

Comment #6. However, in clinical samples, it is approximately 0.1%. What does this figure (0.1%) refer to?

Response:

The theoretical sensitivity of NGS is said to be 0.05-2%, but in clinical samples the sensitivity is said to be around 0.1% due to background errors and other factors. We thought this would not meet the required sensitivity. This has been described in the Introduction (line 64 on page 2 - line 65 on page 2).

Methods:

DNA/RNA was extracted from BC cell lines.

Also, FFPE samples from 212 patients have been utilised. ER+ BC has been mentioned here but this has not been sufficiently characterised and introduced in the introduction section.

Again, the authors assume reader knowledge of various hormone receptor statuses in table 1.

Ethical approval information has been provided which is great.

2.3. mRNA extraction from FFPE

Comment #7. Why did the authors only test these mutations Y537S (1610A>C) and D538G (1613A>G)? Again, the introduction did not sufficiently introduce rare and common mutations of ESR1 and evidence supporting the reported frequency of any of these.

Response: ESR1 mutations were analyzed for the two most representative locations, Y537S (1610A>C) and D538G (1613A>G). These mutations are the most frequent sites of ESR1 mutations. This has been described in the Introduction (line 76 on page 2 - line 78 on page 2).

Results:

Comment #8. The ESR1 expression mRNA/ DNA was measured in 10 cell lines, please explain in more detail why this ratio was necessary to establish.

Response:

ESR1 expression levels between DNA and mRNA were compared in 10 breast cancer cell lines, to verify that mRNA provides a higher copy number in ER positive Breast cancer. We added the text in the Results (line 145 on page 4).

3.2.

Comment #9. Please check your supplementary files since I can only see a file with oligo/ primer information and cannot see figure Ss and have not been able to evaluate it.

Response:

I apologize for forgetting to attach the supplementary figures. We attached Supplementary figures.

3.3.

Comment #10. The authors could add some content on the LNA-clamp method. Also, how was their positive control (DNA of Y537S mutation) established? Where was this known DNA obtained from and how was it validated? Also, was the other mutation also validated using this standardised method? D538G? If not, why not? If possible please add this information.

Response:  

 Y537S and D538G DNA oligo (FASMAC, Kanagawa, Japan) were used as positive controls. We confirmed that these oligos worked with the primers and probes we used. Each mutant-oligos are shown in Table S1. This has been described in the Materials and Methods (line 118 on page 3 - line 121 on page 3).

Comment #11. Figure 1. Sensitivity test of LNA-clamp PCR. Comparison of the sensitivity with serially diluted Y537S mutant DNA (1, 3, 10, 33, and 100 copies) in wild-type DNA (clamp and no clamp, 100,000 copies; reference, 10,000 copies).  The latter bracket is confusing please revise the wording and use “respectively” where appropriate so the various copy numbers of each condition are easier to follow.

Response:

As the reviewer suggested, we corrected as follows,

“in wild-type DNA (with or without clamp and reference are 100,000 copies and 10,000 copies respectively).” This has been described in the Figure 1 (line 175 on page 5 - line 176 on page 5).

Comment #12. Again, I can’t check Figure S4 since they don’t appear in my downloaded supplementary files, the title below is the only thing that I can see.

Response:

We apologize for this. We attached supplementary figures.

Table S1. Primers, probes, and LNA-oligos for ddPCR. 

Comment #13. In lines 160-162, can the authors provide information on between patient DNA copy normalisation? How did the authors standardise their samples for various patients? From Table 3 I can see that different copy numbers were used for each patient. Were any efforts made to dilute each patient sample and measure VAFs to see if the percentage holds?

Response:

Regarding to the standardization of the samples, we first checked the copy number of the reverse-transcribed cDNA ddPCR by 1μL of extracted DNA, and we adjusted input copy number to 100,000 copies for LNA-clamp ddPCR. This has been described in the Materials and Methods (line 132 on page 4 - line 135 on page 4).

Since the detected mutation is a low VAF, just on the edge of sensitivity, we did not dilute the sample to confirm the mutation.

3.4. 

In lines 169-171, the authors have provided the count of patient samples showing the two mutations and have calculated the VAF in the associated table which is good.

Comment #14. I was going to suggest that the authors add indicate details about the 212-patient mutation status but since only 2 mutations were studied, table 3 summarises this information. However, later in the manuscript, the authors talk about therapy status so you could give more information about this in a table.

Response:

We added the Table about ESR1 mutation and adjuvant AI treatment (Table S2). But we think it is nonsense to examine the relationship between ESR1 mutation status and whether or not AI was used for adjuvant therapy, because this study was the retrospective study. On the other hand, regarding only 13 patients with recurrence, the ESR1 mutation positive group tended to have a shorter duration of response to AI treatment for recurrence, although this difference was not statistically significant (Table S3,4). This suggests resistance to AI due to clonal selection, but the results are based on a very small number of cases. This has been described in the Results (line 204 on page 5 - line 207 on page 5).

Comment #15. The authors have performed statistical analysis to compare the clinicopathological features between the larger group 212 and those of the 2-ESR1 mutation group. What type of characteristics has been enriched in the latter group (please add this information to the results section in line 174)? The authors haven’t explained this finding at all. Did any of the patients show both mutations, if not why?

Response:

There were no factors that differed significantly between the two groups with or without mutation.

Double mutation was observed in one case. This has been described in the Results (line 199 on page 5 - line 200 on page 5).

Comment #16. It would be useful if the authors could draw some comparisons with patients' post-aromatase therapies and if/how the VAFs of these two mutations have been altered. This will give some preliminary clues to the truncal/ clonal evolution of these genetic alterations and how treatment may affect them. This could be a comparison to other published studies. The authors could easily perform some statistical analysis to compare them. This is particularly a question since they mention this point in their introduction.

Response:

We also thought it would be very useful if we could compare the mutations before and after AI treatment, but only 12 of the 27 cases with mutations had been treated with AI. Unfortunately, we could not analyze those 12 cases either, because we could not obtain tissue samples after AI treatment. In the future, we plan to examine the comparison of mutations before and after treatment.

Comment #17. Also, with respect to the other patients who did not show these 2 mutations, was this a copy number/ sensitivity issue rather than a 0/1 absent/present issue? Studies have suggested that many primary tumors carry resistance genes in low copies (the point the authors refer to). If so, what method do the authors suggest the scientific community could use to find these ultra-low copy mutations?

Response:

As the reviewer pointed out, there were cases in which sufficient concentrations were not obtained and copy numbers were insufficient (12 cases with less than 10,000 copies), and it is possible that some of these cases included those that were not detected due to insufficient sensitivity. But there is no method that is more sensitive than this method at present.

Comment #18. In this study, no significant difference was found in prognosis according to the presence or absence of ESR1 mutations. I assume the authors refer to the bottom of Table 3. I am confused about what they mean since the p-value is significant.

Response:

We set p< 0.05 as a significant difference, so there were no factors in the table3 that were below this level. This was noted in the Statistics section.

Comment #19. This is thought that only 12 (44%) of the 27 patients with ESR1 mutations received AI as adjuvant therapy.  This is news to me since I thought that all the patients were pre-treatment and this was not discussed in the manuscript. Please add sections to the manuscript to outline pre-treatment and post-treatment patients in this study and add plots to show the distinction between these two groups with respect to treatment status also update the tables with this information.  Also, what is meant by it is thought? Treatment status is not a matter of contention, 12/27 patients have either received treatment or they haven’t, kindly alter this sentence.

Response:

We apologize for the misunderstanding. The analyzed samples were all pre-treatment primary samples. We added the following sentence to the Materials and Methods “No patients received preoperative therapy before tissue sampling.” (line 100 on page 3 - line 101 on page 3). Regarding as the treatment, we added the Tables S2,3,4. The presence or absence of ESR1 mutations did not influence the choice of adjuvant treatment because this study was a retrospective study. When we discussed only 13 recurrent cases, the ESR1 mutation positive group tended to have a shorter response time to AI treatment for recurrence (not statistically significant). This suggests resistance to AI due to clonal selection, but these results are based on very few cases. A larger number of cases would make more difference to this result. This has been described in the Discussion (line 244 on page 7 - line 248 on page 7).

We also added the following sentence to the Discussion “One of the reasons for this is that only 12 (44%) of the 27 patients with ESR1 mutations in primary tumors received AI as adjuvant therapy. Even if the primary tumor has an ESR1 mutation as a minor clone, it has no effect on the prognosis when clonal selection could be avoided.” (line 241 on page 7 - line 244 on page 7).

Comment #20. Second, recurrent cases were limited; thus, we could not confirm the same ESR1 mutation in recurrent tissues. The authors should give more detail about which samples were recurrent. I thought all samples used were primary (primary meaning primary tumour as opposed to patient tissue). How many of your samples represented recurrent disease or secondary tumours?

Response:

We apologize for the misunderstanding as well. All samples analyzed were primary tumor samples before treatment. We followed the treatment history and prognosis since the sample collection as a retrospective observational study.

We modified the limitation as follows “Second, because of the limited number of cases that recurred, it was not possible to trace the mutations found in the primary tumor in the recurrent tissue.” (line 252 on page 7 - line 254 on page 7)

Comment #21. From the conclusion I am not sure if the authors felt that their manuscript is more of a methods article or a patient mutation article.

Response:

We have established LNA-clamp PCR as a method for analyzing rare mutations, so we intend to publish this paper as methodology. We used this method to analyze mutations in ESR1 because ESR1 is a gene that was likely to influence treatment choice in the future by detecting the minor clone in the primary tumor.

Round 2

Reviewer 1 Report

Revisions addressed all the previous comments.

Reviewer 2 Report

The authors have addressed my comments.